# Outcome of induction and associated factors among induced labours in public Hospitals of Harari Regional State, Eastern Ethiopia: A two years' retrospective analysis

Yimer Mohammed Beshir[1], Mohammed Abdurke Kure[2]*, Gudina Egata[3], Kedir Teji Roba[2]

1 Department of Midwifery, Hiwot Fana Specialized University Hospital, Haramaya University, Harar, Ethiopia, 2 School of Public Health, College of Health and Medical Sciences, Haramaya University, Harar, Ethiopia, 3 School of Nursing and Midwifery, College of Health and Medical Sciences, Haramaya University, Harar, Ethiopia

* mameelemo@gmail.com

**Data Availability Statement:** All relevant data are within the paper and its Supporting information files.

## Abstract

### Background

Induction of labor (IOL) is an essential intervention to reduce adverse maternal and neonatal outcomes. It is also improved pregnancy outcomes, especially in resource-limited countries, where maternal and perinatal mortality is unacceptably high. However, there is a scarcity of evidence regarding the outcome of induction of labor and its predictors in low-income countries like Sub-Saharan Africa. Therefore, this study was aimed at assessing the outcome of induction of labor and associated factors among mothers who underwent labor induction in public Hospitals of Harari Regional State, Estern Ethiopia.

### Methods

A facility-based cross-sectional study was conducted from 1 to 30 March, 2019 in Harari Regional State, Eastern Ethiopia. A total of 717 mothers who underwent induction of labor in public Hospitals of Harari Regional State, Eastern Ethiopia from January 2017 to December 2018 were enrolled in the study. Data were collected using a pretested structured questionnaire. The collected data were entered into Epi-data version 3.1 and exported to SPSS version 24 (IBM SPSS Statistics, 2016) for further analysis. A multivariable logistic regression analysis was performed to estimate the effects of each predictor variable on the outcome of induction of labor after controlling for potential confounders. Statistical significance was declared at p-value <0.05.

### Results

Overall, the prevalence of success of induction of labor was 65% [95% CI (61.5, 68.5)]. Pre-eclampsia/eclampsia was found to be the most common indication for induction of labor (46.70%) followed by pre-labor rupture of fetal membrane (33.5%). In the final model of

**Funding:** This study was funded by Haramaya University. The funders had no role in a study design, data collection and analysis, decision to publish, or preparation of the manuscript.

**Competing interests:** The authors have declared that no competing interests exist.

multivariable analysis, predictors such as: maternal age < 24 years old [AOR = 1.93, 95%CI (1.14, 3.26)], nulliparity[AOR = 0.34, 95%CI(0.19, 0.59)], unfavorable Bishop score [AOR = 0.06, 95%CI(0.03, 0.12)], intermediate Bishop score [AOR = 0.08, 95%CI(0.04, 0.14)], misoprostol only method [AOR = 2.29, 95%CI(1.01, 5.19)], nonreassuring fetal heart beat pattern [AOR = 0.14, 95%CI (0.07, 0.25)] and Birth weight 3500 grams and above[AOR = 0.32, 95% CI (0.17, 0.59)] were statistically associated with the successful outcome of induction of labor.

## Conclusion

The prevalence of successful of induction of labor was relatively low in this study area because only two-thirds of the mothers who underwent induction of labor had a successful of induction. Therefore, this result calls for all stakeholders to give more emphasis on locally available induction protocols and guidelines. In addition, pre-induction conditions must be taken into consideration to avoid unwanted effect of failed induction of labour.

## Introduction

Induction of labor (IOL) is an artificial initiation of labour or uterine contraction before the onset of spontaneous true labour in situations when the benefits of delivery of the fetus are outweighing the continuity of the pregnancy [1, 2]. The main aim of initiation of labor without its true time is to have the health of the mother and unborn fetus and to minimize severe obstetric complications related to unnecessary cesarean section [1]. However, sometimes this artificial initiation of labour may result in failure of achieving a good uterine contractions leading to failed induction. This failed IOL is associated with an increased risk of numerous adverse maternal and perinatal outcomes [3, 4].

Globally, the prevalence of success of induction of labor varies across the continents. A secondary analysis of World Health Organization (WHO) on the outcomes of induction of labor in sixteen Asian and African countries indicated the average success of IOL in Asian countries was 81.6%, ranging from 67.9% in China to 90.10% in Cambodia, and 83.4% in African countries ranging from 57.3% in Uganda to the highest range 95% in Angola [5]. Besides, the analysis showed the most common type of induction method in both continents was oxytocin only method which accounts an average proportion of 86.1% and 82.4% in African and Asian countries respectively [5]. In Ethiopia, the success of IOL was lower than the average rate of Asian and African countries. For instance; a previous few studies conducted in Ethiopia have shown a success of induction of labor as 65.7% in Jimma, Northern Ethiopia [6], 61.6% in Hawassa, Southern Ethiopia [7], and 62.2% in Addis Ababa Army referral Hospital, central Ethiopia [8].

Although induction of labor for termination of pregnancy is acceptable and effective, sometimes it has adverse consequences on the health of the mother and unborn fetus [9]. For instance; a previous studies have shown that the effects of outcomes of IOL are not only on the modes of delivery but also causes adverse maternal and perinatal outcomes such as postpartum hemorrhage [10], hyperstimulation of the uterus that results in uterine rupture, chorioamnionitis, endometritis [11], fetal hypoxia, maternal fluid intoxication [12], stillbirth [13] and severe birth asphyxia [14].

Furthermore, researchers have found that post-term pregnancy, hypertensive disorders (pre-eclampsia/eclampsia) during pregnancy, pre-labor rupture of membrane (PROM) [15],

post-term pregnancy [16], intrauterine growth restriction(IUGR), intrauterine fetal death (IUFD) [3], abruption placenta, fetal congenital anomalies [9], and other medical disorders are some indications for the intervention of induction of labor and may influence the success of induction of labor [9, 17]. Moreover, studies have shown that several factors are associated with the success of IOL to achieve vaginal delivery. This success rate IOL can affected by factors such as methods of induction of labour, methods of cervical ripening either by surgical methods (artificial rapture of the membrane, balloon catheter, laminar) or pharmacological methods (Oxytocin, misoprostol) [18–20]. For example; factors like maternal age, gestational age of a pregnancy [21–23], multiparity [4], birth weight of less than 3500gm [24], and favorable cervical status [25] increase the likelihood of success rate of induction of labour.

In Ethiopia, although rare studies have been identified with different study types, still there is a scarcity of locally generated evidence regarding outcome of induction of labour and its predictors in Eastern Ethiopia [6, 8]. In Ethiopia, although few studies have been conducted in the last five years, almost all previous researchers were selective to central and Northern parts of the country(Addis Ababa, Amhara, and Tigray regions) [26–28], rarely to the Southern and Oromia regions [15, 29], and neglecting other parts of the country, particularly Afar, Harari, and Somali regions. Furthermore, to the best knowledge of the researchers, the outcomes of induction of labor has not been studied in this study area. Therefore, this study was aimed to fill this gap by assessing the outcome of induction of labor and its associated factors among mothers who underwent labor induction in public Hospitals of Harari Regional State, Eastern Ethiopia.

## Methods and materials

### Study setting, period, and design

A facility-based cross-sectional study (retrospective chart review) was conducted from 1 to 30 March, 2019 in two public Hospitals (Jugal Regional Hospital, and Hiwot Fana Specialized University Hospital) found in Harari Regional State, Eastern Ethiopia. Harari Region is one of the nine regional states of the Federal Democratic Republic of Ethiopia and covers an area of 334km$^2$. The Region is located in the Eastern part of Ethiopia at a distance of 526 km from the capital, Addis Ababa. According to the 2007 census conducted by the Central Statistics Agency (CSA) of Ethiopia, the total population of the Region was 183,415 (92,316 males and 91, 099 females) [30]. In the Region, there are 45 health facilities (34 health posts, 8 health centers, and 5 hospitals). Among the 5 hospitals found in the Regional State, only two of them are giving service as public hospitals. This study was conducted in these public hospitals, where different and multidimensional health care services being provided to the patient.

### Population and sampling technique

All randomly selected women who delivered after induction of labor, and whose gestational age of pregnancy was greater than or equal to 28weeks in the selected public hospitals of Harari Regional State from January 2017 to December 2018 were enrolled in the study. However, mothers' medical records/charts with incomplete documentation and lacking pertinent information were excluded from the study. In this study, the sample size was determined by considering different factors associated with the outcome of induction of labor using EPI-Info version 7.0 (USA, 2016). Thus, from the predictor variables, maternal parity was considered because it produces a maximum sample size. Thus, we took a previous research report from Suadi Arabia [31]. In this regard, we considered nulliparity as exposure because we follow the assumption that the success of induction of labor would be better in multiparous women than nulliparous. Thus, the success of induction of labor among nulliparous was considered as exposed and the success of induction of labor among multiparous women was considered as

unexposed. Based on this information, the following assumptions were made. Proportion of outcome among nulliparous (p = 59%), the proportion of outcome among multiparous (p = 47.8%), two-sided confidence level = 95%, a tolerable margin = 5%, power of 80%, the ratio of unexposed to expose = 1.0 and by adding 10% contingency for non-response rate, the final sample size for the study was 726.

## Sampling techniques and procedures

Initially, the Harari Regional State was selected purposely as the study site. In the Harari Region, there are only two public Hospitals (Jugal Regional Hospital (JRH) and Hiwot Fana Specialized University Hospital (HFSUH)). HFSUH is the only referral Hospital hosted by Haramaya University in Eastern Ethiopia. Currently, both hospitals are providing different health services including labor and delivery services to the laboring mothers and newborns in maternity wards. According to the health management and information system (HMIS), the annual numbers of delivery reports were 2463 in Jugal Regional Hospital and 4256 in Hiwot Fana Specialized University Hospital. Similarly, previous information obtained from the hospitals' delivery and discharge registration logbooks revealed that the annual number of women who managed with the induction of labor in 2018 was 307 in Jugal Regional Hospital and 552 in Hiwot Specialized University Hospital. Based on the above information, we reviewed two years data of the Hospitals' registration logbooks of the labor ward. Thus, a total of 628 and 1134 mothers who underwent IOL at the gestational age of 28 weeks and more in JRH and HFSUH from January 2017 to December 2018 were identified respectively. We used a simple random sampling (SRS) technique to select study participants using a list frame of the delivery registar. The total sample size (n = 726) was proportionally allocated to both Hospitals. Accordingly, 256 charts were allocated for Jugal Regional Hospital and 470 charts were allocated for Hiwot Fana Specialized University Hospital. Finally, the patients' charts were retrieved and pertinent information was obtained until the required sample size was achieved (Fig 1).

## Data collection tools and procedures

Data were collected through a review of medical records using a structured questionnaire, which was prepared and customized after reviewing different relevant kinds of literature [6, 15, 26]. Clients' charts, labor ward logbooks, discharge logbooks, and operation room logbooks were reviewed to collect the required data. Data were collected by six diploma Midwives who were trained on data collection tools and procedures under the supervision of three Bachelor of Science (BSc) nurses, and the principal investigator. Informed voluntary consent was sought from all authorized bodies of the Hospitals. All eligible medical records were manually and exhaustively searched from where they previously stored and filed in the board cabinets. Eligible charts were searched and allocated using patients' Medical Record Numbers (MRNs) from the listed frames. All collected data were reviewed and checked by supervisor and the principal investigator for completeness, consistency, and if any missing blanks was found, corrective measures were taken immediately.

## Study variables and measurements

In this study, the dependent variable was the outcome of induction of labor. This dependent variable was dichotomized into binary outcomes as 0 and 1. Thus, the successful of induction of labor was recoded as 1 and failed of induction of labor was recoded as 0. **Success of induction of labor is** defined as if a woman delivered vaginally either spontaneously or by instrument after induction **and failed induction** can be defined a**s** if a woman delivered by cesarean section(C/S) due to failure to acquire either adequate uterine contraction ($\geq$3 contractions or

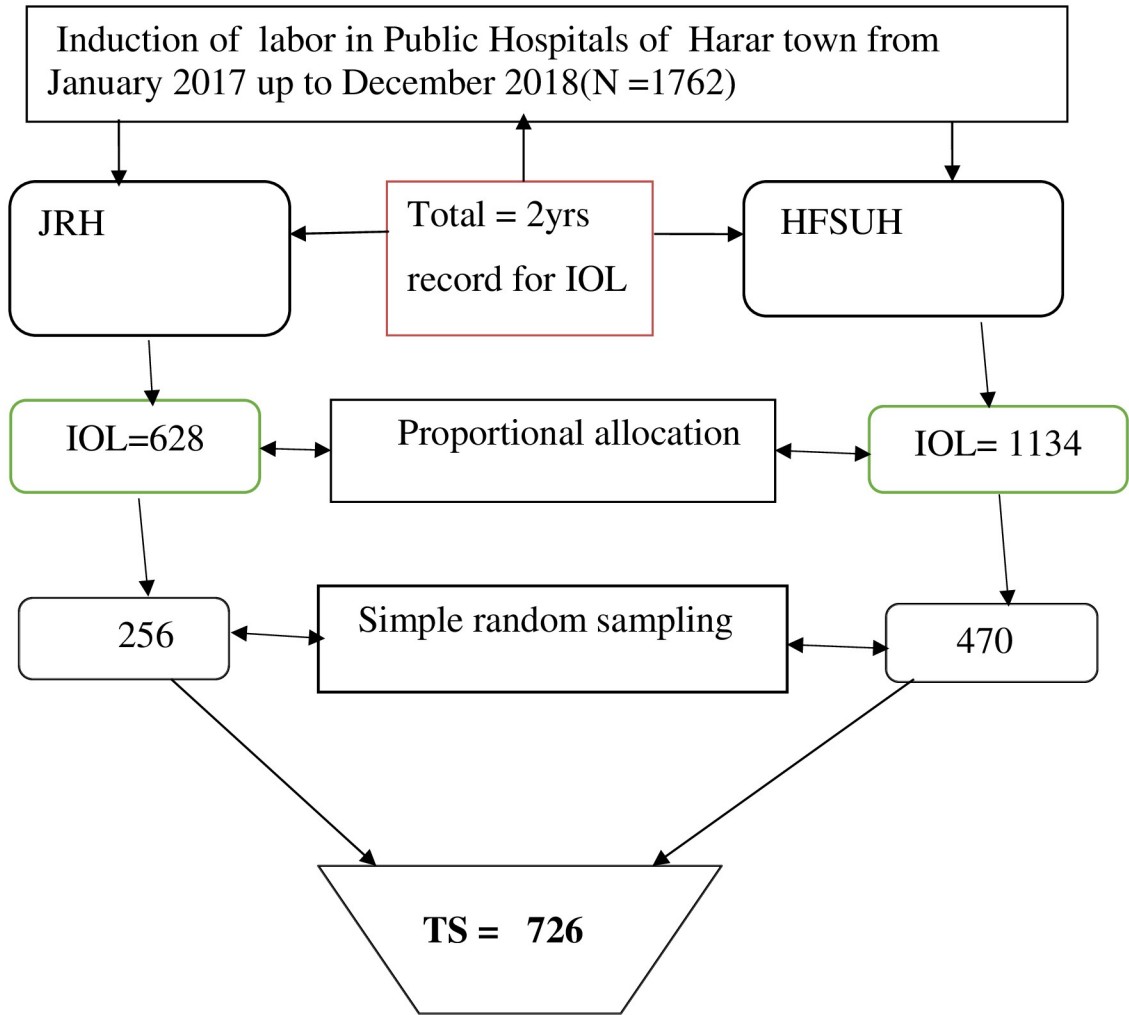

**Fig 1. Schematic presentation of sampling procedure among mothers who delivered after induction of labor in public Hospitals of Harari Regional State, Eastern Ethiopia, 2019.**

contractions lasting ≥40 seconds in 10 minutes period) or failed to show favorable cervical changes (reach at least 4cm in dilatation and fully effaced) despite being on oxytocin drip for at least eight hours or diagnosed as failed induction by any indications for any indication of cesarean section or it is diagnosed if adequate uterine contractions are not achieved after 6 to 8 hours of oxytocin administration and use of the maximum dose for at least one hours [6, 18].

The explanatory variables were categorized as socio-demographic factors (age and residency), obstetrics related factors (ANC follow-up, previous bad obstetrics history, gravity and parity, gestational age status, and Bishop Score status), obstetric indications related factors ((preeclampsia/eclampsia, post-term pregnancy, the pre-labor rapture of the membrane (PROM), polyhydramnios, oligohydramnios, abruptio placenta, and intrauterine fetal death (IUFD)), fetal-related factors(nonreassuring fetal heartbeat pattern (NRFHBP), meconium passage, fetal weight, and fetal sex), methods of IOL related factors (Induction methods (oxytocin only, misoprostol only, oxytocin and artificial rupture of the membrane (ARM)), the time interval from initiation of induction till delivery of the fetus, and type of induction (elective and emergency)).

## Measurements

**Bishop Score System**: The Bishop Score predicts the likelihood of vaginal delivery after induction with Oxytocin. With this scoring system, a number ranging from 0–13 is given to rate the condition of the cervix and fetal station. Interpretation of the Bishop's score: **Score<4**: Unfavorable cervix is unlikely to yield for induction; cervical ripening is needed for success with induction. Postpone induction for next week if possible or use cervical ripening and plan induction for the next day. **Score 5–8**: Intermediate, **Score = 9**: Favorable cervical condition and induction is likely to succeed and there is no need for cervical ripening. Induction using Oxytocin can be planned for the next day [2, 18]. **Oxytocin dosage protocol**: According to Ethiopia FMOH, the national induction protocol was adopted and modified based on the WHO recommendation for induction of labor. According to this protocol, the oxytocin dosage is given in three doses for both primigravida and multigravida. This was identified in supplementary file one (S1 File).

## Data quality control

A structured checklist, which consists of five sections, was prepared in the English language. Data collectors along with the supervisors were trained for one-day. The training was conducted regarding purpose of the study, data collection tool, data collection procedures, and data handling. A pretest was conducted on 30 mothers' records (5% of the total sample) to ensure the validity of the tool, and the correction was made before the actual data collection. The principal investigator and supervisors checked on the spot and reviewed the checklists to ensure completeness and consistency of the information and immediate action was taken accordingly. Double data entry was done by two data clerks and the consistency of the entered data was cross-checked. Simple frequencies and cross-tabulations were done for missing values and outliers. The crosschecked was undertaken with hard copies of the collected data.

## Data processing and analysis

The collected data were checked, coded, and entered into Epi data version 3.1 to minimize logical errors and design skipping patterns. Then, they were exported to SPSS windows version 24 (IBM SPSS Statistics, 2016) for further analysis. Descriptive statistics were carried out using simple frequency tables, proportions and summary measures. A bi-variable logistic regression analysis was used to identify the association between each independent variable and the outcome variable by using binary logistic regression. All variables having p-value $\leq$ 0.25 in the bivariable analysis were included in the final model of multivariable analysis to control for potential confounders. Multi-collinearity was checked using variance inflation factor (VIF) and tolerance, and no collinearities were detected. Likewise, Hosmer-Lemeshow goodness of fitness test was used to check for model fitness and the result was found to be insignificant ($p$ = 0.489), which indicates the model was well fitted. In the final model of multivariable logistic regression analysis, the Adjusted Odds Ratios (AOR) with 95%CI were estimated to identify the effects of independent variables on the outcome of induction of labor. Level of statistical significance was declared at a p-value <0.05.

## Ethical considerations and consent to participate

Ethical clearance was obtained from Institutional Health Research Ethics Review Committee (IHRERC) of College of Health and Medical Sciences, Haramaya University. Supportive letters were written to Jugal Regional Hospital and Hiwot Fana Specialized University Hospital. All patient data were previously anonymized before consent was sought from the authorized

bodies. Medical records of mothers who underwent labor induction from January 2017 to December 2018 were selected. These medical records of mothers were manually searched, and accessed from March 1st to 30th, 2019. Data confidentiality was maintained through anonymity by removing any personal identifiers. Confidentiality of the patient information was assured by omitting their names and using card numbers instead.

## Results

### Socio-demographic and obstetrics related characteristics of the participants

In this study, a total of 726 records of mothers who underwent induction of labor in selected public hospitals of Harari Regional State were retrieved and 717 charts were successfully extracted making the response rate of 98.80%. Nine charts were excluded from the analysis because of incompleteness and lack of pertinent information. The mean age of the mothers was 24.5 years (SD = ±6.86) ranged from 16 to 44 years. The majority 464 (64.70%) of the study participants were from rural setting. More than two-thirds (508, 70.85%) of the respondents had a history of antenatal care follow-up in the previous pregnancy. The mean gestational age was 36.63 weeks (SD ±3.35) and more than half (51.90%) of the mothers were within 37–40 weeks gestational age of a pregnancy (Table 1).

**Table 1. Socio-demographic and obstetric related characteristics of mothers who delivered after induction of labor in public Hospitals of Harari Regional State, Eastern Ethiopia, 2019.**

| Characteristics | Categories | Frequency (n) | Percentage (%) |
|---|---|---|---|
| Age (years) | 16–19 | 92 | 12.83 |
| | 20–24 | 214 | 29.85 |
| | 25–29 | 198 | 27.62 |
| | 30–34 | 101 | 14.08 |
| | $\geq$35 | 112 | 15.62 |
| Residency | Rural | 464 | 64.70 |
| | Urban | 253 | 35.30 |
| Has ANC follow up | Yes | 508 | 70.85 |
| | No | 209 | 29.15 |
| Gestational age (in weeks) | $\leq$36 | 294 | 41.00 |
| | 37–40 | 372 | 51.88 |
| | $\geq$41 | 51 | 7.12 |
| Parity | Nulli-para | 344 | 47.98 |
| | Primi-para | 82 | 11.44 |
| | Multi-para | 291 | 40.58 |
| Bad obstetrics history | Yes | 107 | 14.92 |
| | No | 610 | 85.08 |
| Pre-induction Bishop score | Unfavorable | 226 | 31.52 |
| | Intermediate | 206 | 28.73 |
| | Favorable | 285 | 39.70 |
| Pre-labor rapture of Membrane | Yes | 240 | 33.50 |
| | No | 477 | 66.50 |
| Non-reassuring fetal heartbeat pattern | Yes | 133 | 18.55 |
| | No | 584 | 81.45 |
| Duration of IOL to till delivery | $\leq$10hrs | 500 | 69.70 |
| | >10hrs | 217 | 30.30 |

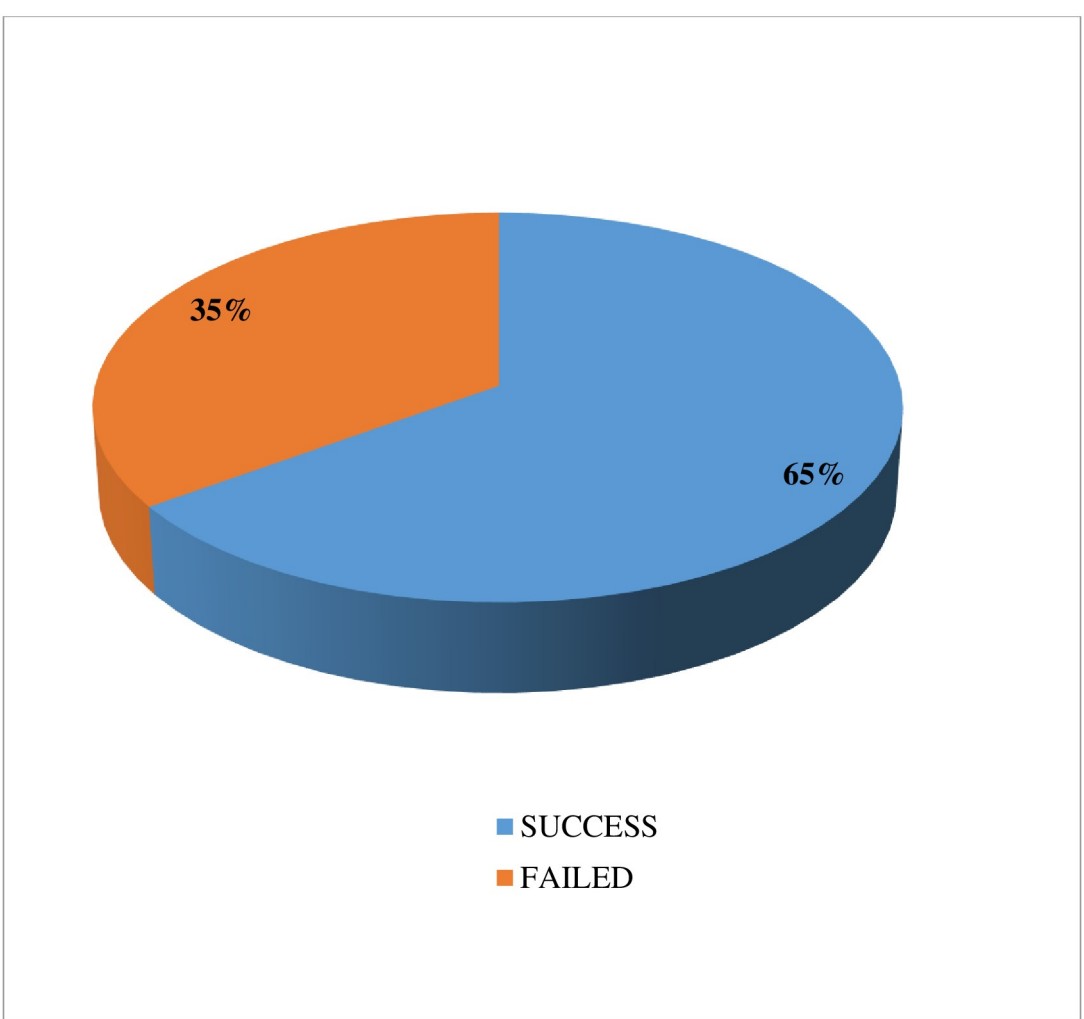

**Fig 2. Magnitude of the outcomes after induction of labor among mothers who delivered after induction of labor in public Hospitals of Harar town, Eastern Ethiopia, 2019.**

### Magnitude of the outcome of induction, and indications of induction

In this study, the proportion of success of induction of labor was 65% [95%CI (61.5,68.6) (Fig 2).

Regarding the indication of induction, the most common indication for induction of labor was pre-eclampsia/eclampsia which accounted for 335(46.70%), followed by pre-labor rupture of membrane 240 (33.50%) and intrauterine fetal death 54(7.5%) (Fig 3).

### Methods of induction of labor, and mode of delivery

Of 717 mothers who underwent labor induction and enrolled in the study, 278(38.77%) of them underwent cervical ripening. Regarding the methods of induction of labor, the most common type of induction method was oxytocin intravenous drip regimen only 633(88.3%) followed by jointly oxytocin and misoprostol regimens 45(6.30%). Concerning the mode of delivery, of the seven-hundred-seventeen study participants included in the study, more than half 411(57.32%) of them gave birth through spontaneous vaginal delivery followed by cesarean section 251(35.01%) and operative vaginal delivery 51(7.67%) (Table 2).

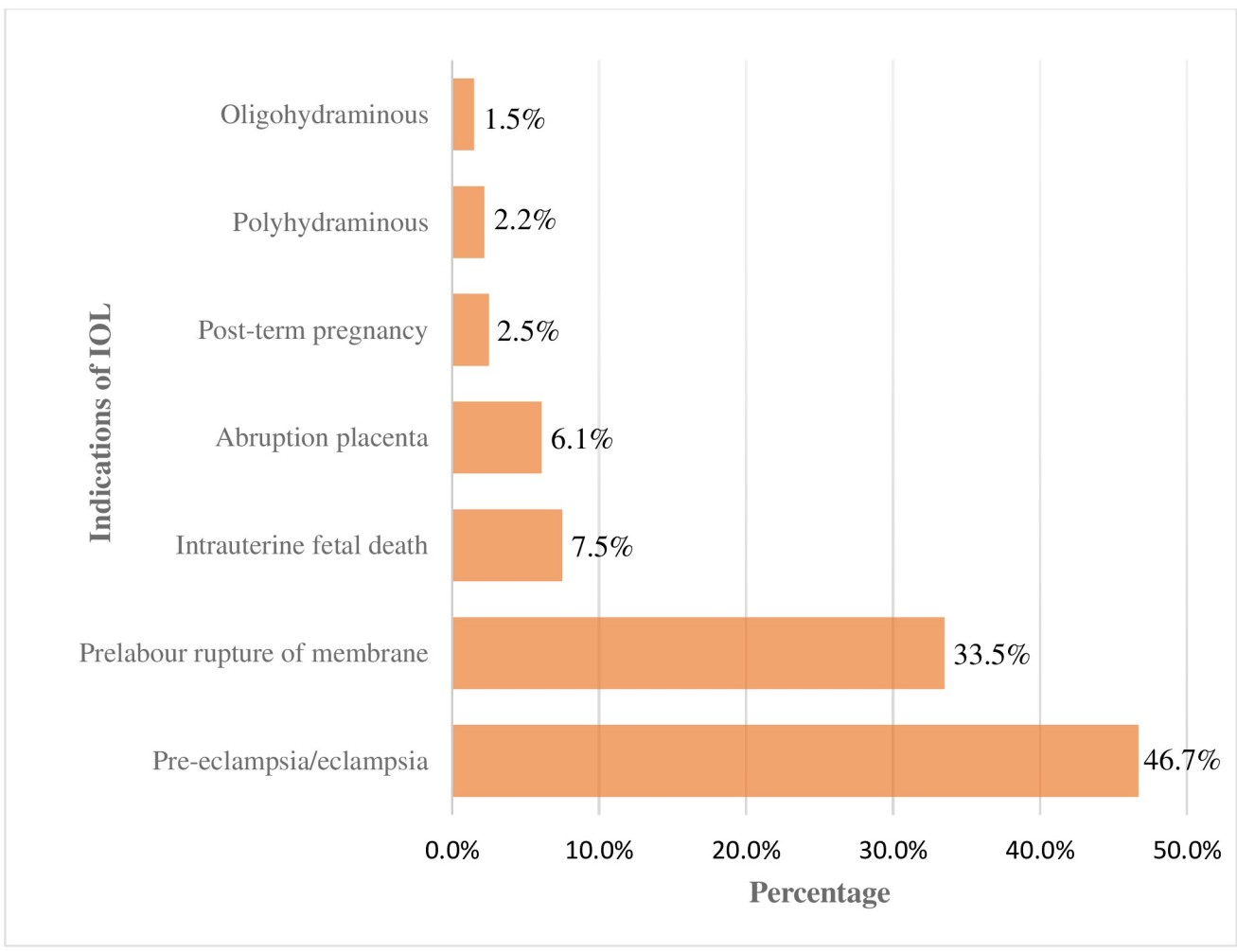

**Fig 3. Indications for induction of labor among mothers who delivered after induction of labor in public Hospitals in Harari Regional, Eastern Ethiopia, 2019.**

### Factors associated with the outcome of induction of labor

In bi-variable logistic regression analysis (age, residency, parity, pre-induction Bishop score, PROM, methods of induction, nonreassuring fetal heartbeat pattern, neonatal weight) were significantly associated with outcomes of induction of labor (Table 3).

In the final model of multivariable logistic regression analysis, variables such as age, parity, pre-induction Bishop score, methods of induction of labor, nonreassuring fetal heartbeat pattern, and birth weight of the newborn were remained significantly associated with the outcomes of induction of labor. Accordingly, the odds of successful of induction of labor were two times higher among mothers whose age ≤24 years of age than those whose age was greater than 24years [AOR = 1.96, 95%CI (1.16, 3.31)]. Likewise, the odds of successful of induction of labor were 67% times lower among nullipara mothers compared to their counterparts (multipara women) [AOR = 0.33, 95%CI (0.19, 0.59)]. Moreover, the likelihood of successful of IOL was decreased by 94% among mothers who had unfavorable cervical status than those who had favorable cervix [AOR = 0.06, 95% CI (0.03, 0.12)]. Similarly, the likelihood of successful of induction of labor was 92% times lower among mothers who had intermediate cervical

**Table 2. Methods of induction of labor and mode of delivery among mothers who delivered after induction of labor in public Hospitals, Harari Regional State, Eastern Ethiopia, 2019.**

| Variables | Categories | Frequency (n) | Percentage (%) |
|---|---|---|---|
| Cervical Ripening (N = 717) | Yes | 278 | 38.77 |
| | No | 439 | 61.23 |
| Ripening Methods (N = 278) | Misoprostol | 139 | 50.00 |
| | Dinoprostone | 19 | 6.83 |
| | Balloon Catheter | 120 | 43.17 |
| Methods of IOL (N = 717) | Oxytocin only | 633 | 88.30 |
| | ARM + Oxytocin | 39 | 5.40 |
| | Oxytocin+ misoprostol | 45 | 6.30 |
| Phases of IOL (N = 672) | First | 70 | 10.42 |
| | Second | 103 | 15.33 |
| | Third | 499 | 74.25 |
| Oxytocin drop per Minutes per 1L in Normal saline(N = 672) | 20 | 19 | 2.83 |
| | 40 | 50 | 7.44 |
| | 60 | 119 | 17.71 |
| | 80 | 484 | 72.02 |
| Route for misoprostol(N = 45) | Vaginal | 31 | 68.89 |
| | Oral | 8 | 17.78 |
| | Sublingual | 6 | 13.33 |
| Mode of delivery (N = 717) | SVD | 411 | 57.32 |
| | OVD | 55 | 7.67 |
| | C/S | 251 | 35.01 |

**Key**: ARM- Artificial Rupture of Membrane; SVD: Spontaneous Vaginal delivery; OVD: Operative Vaginal Delivery; C/S: Caesarean Section.

status than those who had a favorable cervix [AOR = 0.08, 95%CI (0.04, 0.14)]. Regarding the method of induction of labor, the odds of successful of IOL were 2.36 times higher among mothers who were induced by jointly oxytocin and misoprostol than those who were induced by oxytocin only method [AOR = 2.36, 95% CI(1.04, 5.32). Likewise, the likelihood of having successful of IOL was 86% times lower among mothers whose fetuses experienced nonreassuring fetal heartbeat patterns than those whose fetuses had no non-reassuring fetal heartbeat pattern [AOR = 0.14, 95% CI(0.07, 0.25)]. In addition, the weight of infant was independently associated with successful of IOL. Thus, the successful of IOL was decreased by 68% among mothers whose fetal weight greater than 3500 grams compared to those whose fetal weight was 25000–3499 grams [AOR = 0.32, 95%CI(0.17, 0.60)] (Table 4).

## Discussion

In this study, the overall proportion of success of induction of labor was found to be 65%. Preeclampsia/eclampsia and prelabour rupture of the membrane were the most common indications for induction of labor. Age of the mother, parity, pre-induction Bishop score, method of induction, nonreassuring fetal heartbeat pattern, and weight of the newborn were factors significantly associated with outcomes of induction of labor.

In this study, the prevalence of success of induction of labor was relatively low. This result is in line with previous studies conducted in Ethiopia like Jimma University Teaching Hospital (65.7%) [6], Hawassa public health facilities(61.56%) [32], and Addis Ababa Army Referral Hospital(62.2%) [8]. Moreover, the finding is also nearly comparable with the cross-sectional

**Table 3. Bivariable logistic regression analysis of factors associated with the outcome of induction of labor among mothers who delivered after IOL in public hospitals of Harari Regional State, Eastern Ethiopia, 2019.**

| Variables | Categories | Outcome of IOL | | COR(95% CI) |
|---|---|---|---|---|
| | | Success (%) | Fail (%) | |
| **Age (Years)** | $\leq$24 | 207(67.60) | 99(32.40) | 1.23(0.9, 1.7)* |
| | >24 | 259(63.00) | 152(37.00) | 1.00 |
| **Residency** | Rural | 319(68.80) | 145(31.20) | 1.58(1.15, 2.18)** |
| | Urban | 148(58.10) | 106(41.90) | 1.00 |
| **ANC follow up** | Yes | 325(64.00) | 183(36.00) | 0.86(0.61, 1.21) |
| | No | 141(67.50) | 68(32.50) | 1.00 |
| **Gestational age** | $\leq$36 | 199(67.70) | 95(32.30) | 1.11(0.8, 1.53) |
| | 37–40 | 243(65.30) | 129(34.70) | 1.00 |
| | $\geq$41 | 24(47.10) | 27(52.90) | 0.47(0.26, 0.85) |
| **Parity** | Nulli-Para | 207(60.20) | 137(39.80) | 0.59(0.42, 0.83)** |
| | Primi-para | 50(61.00) | 32(39.00) | 0.61(0.37, 1.02) |
| | Multi-para | 209(71.80) | 82(28.20) | 1.00 |
| **Bad obstetric history** | Yes | 66(61.70) | 41(38.30) | 0.85(0.55, 1.29) |
| | No | 400(65.60) | 210(334.40) | 1.00 |
| **Pre-induction Bishop score** | Unfavorable | 116(51.30) | 110(48.70) | 0.15(0.10, 0.24)** |
| | Intermediate | 101(49.00) | 105(51.00) | 0.14(0.09, 0.22)** |
| | Favorable | 249(87.40) | 36(12.60) | 1.00 |
| **Pre-labor rupture of fetal membrane** | Yes | 176(73.30) | 64(26.70) | 1.77(1.26,2.49)** |
| | No | 290(60.80) | 187(39.20) | 1.00 |
| **Duration of of IOL till delivery** | $\leq$10 hours | 323(64.60) | 177(35.40) | 0.94(0.68, 1.32) |
| | >10 hours | 143(65.90) | 74(34.10) | 1.00 |
| **Methods of Induction** | Oxytocin only | 405(64.00) | 228(36.00) | 1.00 |
| | ARM+oxytocin | 28(71.80) | 11(28.20) | 1.43(0.70, 2.93)* |
| | Oxytocin+Misoprostol | 33(73.30) | 12(26.70) | 1.55(0.78, 3.06)* |
| **Nonreassuring fetal heartbeat pattern** | Yes | 35(26.30) | 98(73.70) | 0.13(0.08, 0.19)** |
| | No | 431(73.80) | 153(26.20) | 1.00 |
| **Birth weight of the newborn** | <2500 | 158(70.90) | 65(29.10) | 1.18(0.83, 1.69) |
| | 2500–3499 | 265(67.30) | 129(32.70) | 1.00 |
| | $\geq$3500 | 41(42.30) | 56(57.70) | 0.36(0.23, 0.56)** |

Key: ARM-Artificial Rupture of Membrane, IOL-Induction of Labor;

* = p-value < 0.25,

** = statistically significant.

studies conducted in South Africa (59.76%) [17], and Nigeria (63.5%) [16]. The possible justification for these similarities might be because of the definition of time interval for failed IOL (from six to eight hours) for intravenous oxytocin drips. Another possible explanation might be because using a similar standard protocol for induction of labor as the majority of the reports were from similar settings (limited resource countries). However, the current study finding is lower than the studies conducted in different countries like Ethiopia (80.3%) [26], China(76.9%) [33], Pakistan (82%) [34], Saudi Arabia(84%) [31], India (86.32%) [35] and Nigeria (82.2%) [36]. The possible justification for this discrepancy might be due to differences in socio-demographic characteristics of the study participants, the nature of study designs, and methods of data collections techniques. The other differences in estimate are due to the time gap between study periods, the geographical setting of the study population, and the difference

**Table 4. Multivariable logistic regression analysis of factors associated with the success of induction of labor among mothers who delivered after induction of labor in public Hospitals of Harari Regional State, Eastern Ethiopia, 2019.**

| Variables | Categories | Outcome of IOL | | COR(95%CI) | AOR (95%CI) |
|---|---|---|---|---|---|
| | | Success (%) | Fail (%) | | |
| **Age (years)** | ≤24 | 207(67.60) | 99(32.40) | 1.23(0.90, 1.70) | 1.96(1.16, 3.31)* |
| | >24 | 259(63.00) | 152(37.00) | 1.00 | 1.0 |
| **Residence** | Rural | 319(68.80) | 145(31.20) | 1.58(1.15, 2.18) | 1.22(O.86, 1.27) |
| | Urban | 148(58.10) | 106(41.90) | 1.00 | 1.0 |
| **Parity** | Nulli-Para | 207(60.20) | 137(39.80) | 0.59(0.42, 0.80) | 0.33(0.19, 0.59)** |
| | Primi-para | 50(61.00) | 32(39.00) | 0.61(0.37, 1.02) | 0.51(0.26, 1.02) |
| | Multi-para | 209(71.80) | 82(28.20) | 1.00 | 1.0 |
| **Pre-induction Bishop score** | Unfavorable | 116(51.30) | 110(48.70) | 0.15(0.10, 0.24) | 0.06(0.03, 0.12)** |
| | Intermediate | 101(49.00) | 105(51.00) | 0.14(0.09, 0.22) | 0.08(0.04, 0.14)** |
| | Favorable | 249(87.40) | 36(12.60) | 1.00 | 1.0 |
| **PROM** | Yes | 176(73.30) | 64(26.70) | 1.77(1.26, 2.49)** | 1.51(0.92, 2.13) |
| | No | 290(60.80) | 187(39.20) | 1.00 | 1.0 |
| **Methods of Induction** | Oxytocin only | 405(64.00) | 228(36.00) | 1.00 | 1.0 |
| | ARM+oxytocin | 28(71.80) | 11(28.20) | 1.43(0.70, 2.93) | 1.35(0.49, 3.23) |
| | Oxytocin+Misoprostol | 33(73.30) | 12(26.70) | 1.55(0.78, 3.06) | 2.36(1.04, 5.32)*** |
| **NRFHBP** | Yes | 35(26.30) | 98(73.70) | 0.13(0.08, 0.19) | 0.14(0.08, 0.25)*** |
| | No | 431(73.80) | 153(26.20) | 1.00 | 1.0 |
| **Birth weight** | <2500 | 158(70.90) | 65(29.10) | 1.18(0.83, 1.69) | 1.46(0.88, 2.42) |
| | ≥3500 | 41(42.30) | 56(57.70) | 0.36(0.23, 0.56) | 0.32(0.17, 0.60)*** |
| | 2500–3499 | 265(67.30) | 129(32.70) | 1.00 | 1.0 |

Keys: 1 = reference, *p*-values:

** ≤ 0.01,

*** ≤0.001,

COR = Crude Odds Ratio, AOR = Adjusted, NRFHBP = Nonreassuring fetal heartbeat pattern, PROM = Pre-labor rupture of fetal membrane.

in the sample size of the studies. In addition, definitions of failed induction per protocol might be attributed for these observed variations. For instance, type of methods of inductions, cervical ripening methods, maintaining the oxytocin concentration and dose adjustment while changing the infusion of bag might different per protocol.

In the final model of multivariable analysis, maternal age was independently associated with the success of induction IOL. Thus, the odds of successful of IOL were two times higher among mothers less than or equal to 24 years of age than those whose age greater than 24years. This is in line with studies conducted by Batinelli *et al* in Italy and Sara in Addis Ababa, central Ethiopia [4, 8]. This association might be due to maternal anatomical stability in the age group of lower than 25 years old. Reversely, the sacral promontory, ischial spine, and coccyx bone deformity increase (becoming shrink inward to the pelvic cavity that leads to decreasing of pelvic diameter) as age increases. Likewise, the success of induction of labor was affected by maternal parity. Thus, the likelihood having successful of IOL was reduced by more than two-thirds among nulliparous mothers when compared to multiparous women. This finding is also supported by previous studies conducted elsewhere such as King Khalid Hospital in Saudi Arabia, Kenyatta National Hospital in Kenya, and Hawassa Referral Hospital in southern Ethiopia [7, 31, 37]. This is might be due to direct induction before cervical ripening, undoing of cervical sweeping, and amniotomy after the active phase of the first stage of labor in nullipara mothers leads to failed induction. Other possible justifications might be

attributed to multiparity because as the parity of the mother increases, the likelihood of failed induction of labor decreases as uterine muscles can be easily stimulated and contracted in multipara women.

Moreover, pre-induction cervical status (Bishop score) was found to be independent predictor of success of IOL. Accordingly, the successful of labor induction was lower in mothers who had unfavorable and intermediate Bishop scores. Thus, the likelihood of success of induction of labor was greater than ninety-percent among mothers who had a favorable Bishop score compared to those mothers who had unfavorable and intermediate Bishop Scores. This result is supported by studies conducted in Jimma University Specialized Hospital and Hawassa Town Health facilities which reported failed induction of labor in mothers with unfavorable and intermediate Bishop Score [6, 7]. This finding is supported by another study conducted in Addis Ababa, in which higher success of IOL was observed in mothers who had higher cervical Bishop Scores [8]. It is also supported by the scientific finding of different kinds of literature that the condition of the cervix is an important predictor, with the modified Bishop score which is a widely used scoring system that includes four cervical parameters (cervical consistency, effacement, position, and dilatation) and the station of presenting part of the fetus. This indicates that the un-ripened cervix is highly associated with failed induction. This is also might be because un-ripened cervical status (unfavorable cervix) is less likely to be affected by uterine muscle contractility and pressure of the fetal present part compared to the favorable cervix.

In this study, the type of method for induction was found to be an independent predictor of the success of induction of labor. Thus, the odds of successful of IOL were 2.36 times higher among mothers who were induced by jointly oxytocin and misoprostol than those who were induced by oxytocin only method. This result is in line with the previous studies conducted in Brazil, Zimbabwe, and Nigeria [16, 38, 39], which indicated higher success of IOL in combination of oxytocin and misoprostol than using either of the medication a lone. The possible explanation might be attributed to the nature of the regimen used for induction. This is because misoprostol does not need special storage and it is less likely to be affected by high temperature than oxytocin. The difference in the time interval, optimal dose, and mechanism of action on the uterus are also some possibilities.

Furthermore, the presence of fetal heartbeat abnormalities during labor induction was negatively associated with the success rate of induction of labor. Thus, mothers whose fetuses experienced nonreassuring fetal heartbeat patterns were 86% times less likely to have a successful IOL than those whose fetuses had a normal fetal heartbeat pattern. This is in line with the study conducted in South Carolina and in Addis Ababa Military Hospital, central Ethiopia [8, 40]. It is also supported by other studies conducted in Jordan and Ethiopia (Wolliso St. Luke Catholic Hospital), which explained that the absence of fetal heartbeat abnormality had a higher success rate than those who developed abnormal fetal heartbeat pattern [15]. The possible reason might be because the presence of fetal heartbeat abnormalities can cause fetal distress that leads to an increment of failure of induction of labor.

Finally, in this study, the birth weight was independently associated with success of IOL. Accordingly, those mothers whose fetal weight 3500 grams and greater were 68% times less likely to have a success rate IOL than those whose fetal weight was 2500–3499 grams. This finding is supported by studies conducted by Bertinelli *et al* in University Hospital 'Le Scotte' of Siena, Italy and in Kenyatta National Hospital by Rashida [4, 37]. The possible reason could be explained by when the weight of the fetus is greater than 3500 grams it causes cephalo-pelvic disproportion, which leads to the difficulty of vaginal delivery and hence it increases the failure rate of induction of labor.

   

## Limitations of the study

Since we used a chart review cross-sectional study design, no causal association could have made. Moreover, the data were collected from a secondary source; some independent variables might be missed. The study was conducted only in public health institutions; pregnant women who underwent IOL at private health facilities were not included in the study.

## Conclusion

Overall, the proportion of success of induction of labor was relatively low in this study area. Pre-eclampsia/eclampsia and prelabour rupture of the membrane were the most common indications of induction of labor in the study area. Maternal age, maternal parity, Bishop score, method of induction of labor, presence of nonreassuring fetal heartbeat pattern, and newborn weight at birth were independently and significantly associated with the success of induction of labor. Therefore, further work is needed to improve the success of induction of labor by assessing and monitoring maternal and fetal status before the initiation of induction of labor. Moreover, cue due attention should be given to induction protocols and standard guidelines for better outcomes of the success of induction of labor. Moreover, longitudinal studies are needed to identify the causal association of outcomes of induction of labor and its predictors.

## Supporting information

**S1 File. Induction protocol for induction of labour.**
(PDF)

**S1 Data. Data set used for analysis of the study conducted on IOL in Eastern Ethiopia, 2019.**
(ZIP)

## Acknowledgments

The authors thank the data collectors, data collectors' supervisors and administrative staff of Harari Regional Health Bureau for their cooperation to conduct this research paper. We also extend our deepest gratitude to clinical staff of Hiwot Fana Specialized University Hospital and Jugal Hospital for their unreserved support, and without them, this work would not be realized.

## Author Contributions

**Conceptualization:** Yimer Mohammed Beshir, Gudina Egata, Kedir Teji Roba.

**Data curation:** Yimer Mohammed Beshir.

**Formal analysis:** Yimer Mohammed Beshir, Mohammed Abdurke Kure, Kedir Teji Roba.

**Funding acquisition:** Yimer Mohammed Beshir, Gudina Egata, Kedir Teji Roba.

**Investigation:** Yimer Mohammed Beshir.

**Methodology:** Yimer Mohammed Beshir, Mohammed Abdurke Kure, Gudina Egata, Kedir Teji Roba.

**Project administration:** Yimer Mohammed Beshir.

**Resources:** Yimer Mohammed Beshir, Gudina Egata, Kedir Teji Roba.

**Software:** Yimer Mohammed Beshir, Mohammed Abdurke Kure, Kedir Teji Roba.

**Supervision:** Yimer Mohammed Beshir, Gudina Egata, Kedir Teji Roba.

**Validation:** Yimer Mohammed Beshir, Mohammed Abdurke Kure, Gudina Egata, Kedir Teji Roba.

**Visualization:** Yimer Mohammed Beshir, Mohammed Abdurke Kure, Gudina Egata, Kedir Teji Roba.

**Writing – original draft:** Yimer Mohammed Beshir, Mohammed Abdurke Kure, Gudina Egata, Kedir Teji Roba.

**Writing – review & editing:** Yimer Mohammed Beshir, Mohammed Abdurke Kure, Gudina Egata, Kedir Teji Roba.

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
