## [Decision Letter · Decision Letter 0]

17 Jun 2021

PONE-D-21-06515

Outcome of Induction and Associated Factors among Induced Labors in Public Hospitals of Harar Town, Eastern Ethiopia: A two Years’ Retrospective Analysis

PLOS ONE

Dear Dr. Kure,

Thank you for submitting your manuscript to PLOS ONE. After careful consideration, we feel that it has merit but does not fully meet PLOS ONE’s publication criteria as it currently stands. Therefore, we invite you to submit a revised version of the manuscript that addresses the points raised during the review process.

We look forward to receiving your revised manuscript.

Kind regards,

Subash Chandra Gupta, Ph.D.

Academic Editor

PLOS ONE

Journal Requirements:

2. In the ethics statement in the manuscript and in the online submission form, please provide additional information about the patient records used in your retrospective study, including: a) whether all data were fully anonymized before you accessed them; b) the date range (month and year) during which patients' medical records were accessed; c) the date range (month and year) during which patients whose medical records were selected for this study sought treatment. If the ethics committee waived the need for informed consent, or patients provided informed written consent to have data from their medical records used in research, please include this information.

"This study was funded by Haramaya University and the Ethiopian Ministry of Science and Higher

 Education. The funding organizations had no role in the study design, data collection, data

analysis, writing of the manuscript."

"The funders had no role in study design, data collection and analysis, decision to publish, or preparation of the manuscript"

Reviewers' comments:

Reviewer's Responses to Questions

**Comments to the Author**

1. Is the manuscript technically sound, and do the data support the conclusions?

Reviewer #1: Partly

Reviewer #2: Yes

2. Has the statistical analysis been performed appropriately and rigorously? 

Reviewer #1: No

Reviewer #2: Yes

3. Have the authors made all data underlying the findings in their manuscript fully available?

Reviewer #1: Yes

Reviewer #2: Yes

4. Is the manuscript presented in an intelligible fashion and written in standard English?

Reviewer #1: Yes

Reviewer #2: Yes

5. Review Comments to the Author

Reviewer #1: 1.The authors have conducted the research in an ethical fashion and collected all the data using approprate mechanisms to ensure the quality remains appropriate for the subject. The authors have duly cited their peers and prior art and also conducted statistical analysis with rigour. However, the experitmental setup does not indicate how this study is different from the prior studies and therfore provides no indication of why the outcomes might be expected to be different. If the setup was the same the output can be safely assumed to be the same.

2. Based on the current analysis shows no significant improvement in outcomes while utilizing IoL, further analysis is meritted for the use case.

a. The paper seems to describe the effect of the parameters in isolation. The data representation, and therefore results, does not describe the effect of the combination of factors. Authors must conduct this analysis and clearly represent the significance of individual parameters and the various permutations and combinations

b. A detailed description of the control sample (patients not receving IOL) is requried. The current manuscript does not provide details on this.

c. Based on the results provided, the bishops score has a significantly higher effect on the outcome. While this may well turn out to be true, the analysis needs to be provide an analysis while holding the bishops score as a constant. This would allow a better study of the effect of each variable. Without this analysis the effect of the IOL on parameters beyond the bishops score are nebulous at best. A similar analysis should be conducted while holding each parameter constant. This would provide an exhaustive analysis that helps differentiate the output from prior work.

3. Authors should maintain unformity while describing results. Eg: Line 259 describes the paramenter as "more likely" based on the cut off; but the subsequent parameters are described as "less likely" at the cut off. The authors should construct the sentences to describe in one of two ways (either less or more). Switching between positive and negative effect of the cut off can create confusion for the readers.

4. The discussion section focusses more on prior work than an analysis of the results. Perhaps correlation while holding parameters constant would provide more content for discussion.

4. Significant sections of the paper need to be re-written to correct for sentence construction.

5. Suggestion: Some of the tables could be converted to graphical representation for easier interpretation.

Overall the work conducted has potential for significant impact, but a more detailed analysis is needed. In the current fommt the manuscript does not give any indication of differentiation from prior art. However, given the nature of the problem being addressed, I recommend the authors be give a chance to conduct a more details analysis and submit a renewed manuscript.

Reviewer #2: 1 Abstract is written well and comprehensive

2. In the Introduction section, Authors may give more information about advantages and disadvantages of induction of labor (IOL) citing more literature

3. Study design, Study population, Eligibility criteria, and Sample size calculations are well planned and presented; sampling procedure is done well with good selection of variables.

4 The presentation of results obtained is very clear with facts of of many pregnant women having indications of pre-eclampsia/eclampsia and pre-labor rupture of membrane. The state of conditions are worrying in this part Ethiopia reported by the Authors. Parity of the mothers, nullipara and multipara mothers is also taken care in this study

5. The results obtained are discussed well citing references.

6. As an another parameter, in the discussion, the Authors would have touched upon socio-economic and nutritional status of pregnant women compared to those in other countries/Ethiopia.

7. In this study Authors established that success of induction of labor was relatively low in study area

6. PLOS authors have the option to publish the peer review history of their article (what does this mean?). If published, this will include your full peer review and any attached files.

Reviewer #1: **Yes: **Kiran Aatre

Reviewer #2: No

---

## [Author Response · Author response to Decision Letter 0]

29 Aug 2021

Authors’ Response to the editor’s and Reviewers’ comments and Suggestions

Manuscript ID: PONE-D-21-06515

Journal: PLOS ONE

Dear Editors and Reviewers,

Thank you so much for giving us an opportunity to submit a revised draft of our manuscript entitled “Outcome of Induction and Associated Factors among Induced Labors in Public Hospitals of Harar Town, Eastern Ethiopia: A two Years’ Retrospective Analysis” to this high visibility impact factor and peer reviewed Journal. We appreciate the time and effort that you and the reviewers dedicated to providing feedback on our manuscript. We are very grateful for the insightful comments and valuable improvements to our premature paper. We have incorporated most of the suggestions and comments made by handling editor, reviewers. All comments and suggestions are clearly stated and well addressed (a point-by-point to the reviewer’s comments and concerns). These changes are highlighted in Red font color within the clean revised manuscript. 

Authors’ Response to Editor’s Comments and Suggestions

Title: Outcome of Induction and Associated Factors among Induced Labors in Public Hospitals of Harari Regional State, Eastern Ethiopia: A two Years’ Retrospective Analysis

Authors: Yimer Mohammed Beshir, Gudina Egata, Mohammed Abdurke Kure, Kedir Teji Roba, 

To: Handling Editor(s)

From: Mohammed Abdurke Kure (Corresponding Author)

Subject: Submission of Incorporated Comments and Suggestions

First, we thank you for your constructive comments and helpful suggestions that helped us to improve and enrich our manuscript. Here under in the table below, we have pointed out how authors incorporated your valuable comments, suggestions and concerns one by one. 

Editor’s Comments to the Authors

A.Editor’s General Comments and Suggestions 

Authors' Response: Overall, thank you so much for cooperation to handle our manuscript. Handling paper is really needs dedication and strong commitment. Thanks a lot!

1.Thank you for submitting your manuscript to PLOS ONE. After careful consideration, we feel that it has merit but does not fully meet PLOS ONE’s publication criteria as it currently stands. Therefore, we invite you to submit a revised version of the manuscript that addresses the points raised during the review process. 

Authors' Response: Great! We are very happy and overjoyed. Thank you very much for giving us an opportunity to submit our revised manuscript to such legitimate and high visibility impact factor Journal (PLOS ONE). 

Academic Editor’s Specific Comments (Journal Requirements)

2. Please ensure that your manuscript meets PLOS ONE's style requirements, including those for file naming. The PLOS ONE style templates can be found athttps://journals.plos.org/plosone/s/file?id=wjVg/PLOSOne_formatting_sample_main_body.pdf and 

Authors' Response: Thank you so much. You are perfect. These are very important comments. Even it is the authors mandatory to stick to Journal’s format guidelines. Now, the authors addressed this critical issues based on your valuable suggestion. We downloaded all formats templates (PLOS Affiliations Formatting and Manuscript body formatting guidelines) from Journal’s Website, and critically read and corrected all necessary formatting. Newly changed and corrected were highlighted with red font color in the clean revised manuscript.

3. In the ethics statement in the manuscript and in the online submission form, please provide additional information about the patient records used in your retrospective study 

Authors' Response: Thanks. You are perfect. This valid comment was addressed based on your valuable suggestion. Newly changed and corrected was highlighted with red font color in the clean revised manuscript (on page, Lines= )

4.Please remove any funding-related text from the manuscript and let us know how you would like to update your Funding Statement. 

Authors' Response: Ok, thanks. The editor is correct. We are very sorry; this is against PLOS ONE authors’ guideline. The authors critically reviewed this valid comment and corrected the necessary modification. The newly modified change was highlighted with red font color in the clean revised manuscript

5. If there are no restrictions, please upload the minimal anonymized data set necessary to replicate your study findings as either Supporting Information files. We will update your data availability statement on your behalf to reflect the information you provide. 

Authors' Response: Thank you very much. In fact, this is a valid concern. We critically considered this point, and we decided to upload data set used for analysis with revised submission as supplementary File 2(S2)

Authors' Response: Thank you a lot for such a technical input. Now, authors considered the raised issue. We are very sorry! In fact, this authors responsibility to follow and adhere Journal’s format guidelines. After thoroughly and critically revised this important comment, we removed from previously it appeared in Acknowledgment part. Thanks!

End of authors’ responses for Handling Editor(s)

A. Authors’ Response to Reviewer 1’s Comments and Suggestions

Title: Outcome of Induction and Associated Factors among Induced Labors in Public Hospitals of Harari Regional State, Eastern Ethiopia: A two Years’ Retrospective Analysis

Authors: Yimer Mohammed Beshir, Gudina Egata, Mohammed Abdurke Kure, Kedir Teji Roba

To: Reviewer 1

From: Mohammed Abdurke Kure (Corresponding Author)

Subject: Submission of Incorporated Comments and Suggestions

First and foremost, we would like to thank you for your constructive and valuable comments and helpful suggestions that helped us to improve and enrich our premature manuscript. Here under in the table below, we have pointed out how authors incorporated your valuable comments, suggestions and concerns one-by-one. 

Reviewer’s Comments to the Authors 

Reviewer’s General Comments 

1. Is the manuscript technically sound, and do the data support the conclusions?

Reviewer #1: partly 

Authors' Response: Thank you very much. We thank you for your appreciation and constructive suggestion. Further, we revised and enriched the paper after previous initial submission to the journal.

2. Has the statistical analysis been performed appropriately and rigorously?

Reviewer #1: No 

Authors' Response: Thank you so much. Authors acknowledge you for your countless effort. We also appreciate this valuable response. Dear, reviewer, in this study, the authors followed standard methods of data collections and analysis. We used validated data collection tool, which was developed and customized from Ethiopian Demographic and Health Survey (EDHS) data collection tool, and by reviewing related published literature. We also used strong statistical software (Epi-Data version-3.1) for data entry. Further, data were exported to SPSS version 24 (IBM SPSS Statistics, 2016) for further analysis. Now, we revised the paper after initial submission. We took a long time, thoroughly revised and corrected the whole parts of our manuscript to make it more scientifically robust. Further, we also critically revised the statistical analysis of all descriptive statistics, Bi-variable and multivariable logistic regression models to correct any systematic errors introduced during analysis sage. 

3. Have the authors made all data underlying the findings in their manuscript fully available?

Reviewer #1: Yes 

Authors' Response: Thanks a lot! Authors acknowledged your countless efforts. Now, authors agreed to upload SPSS data set used for analysis as supplementary file 2(S2) with revised submission. 

4. Is the manuscript presented in an intelligible fashion and written in standard English?

Reviewer #1: Yes 

Authors' Response: Thank you very much. Even after initially submission, we thoroughly revised and edited the whole parts of our manuscript and extensively corrected all copy-editing errors in the clean revised manuscript. Authors also sent the manuscript to language expert/editor who critically reviewed, edited and corrected all language related errors made in a submitted manuscript.

5. Specific Review’s Comments to the Authors 

1.The authors have conducted the research in an ethical fashion and collected all the data using appropriate mechanisms to ensure the quality remains appropriate for the subject. The authors have duly cited their peers and prior art and also conducted statistical analysis with rigour. However, the experimental setup does not indicate how this study is different from the prior studies and therefore provides no indication of why the outcomes might be expected to be different. If the setup was the same the output can be safely assumed to be the same.

Authors' Response: Thank you very much. We would like to thank you for your appreciation and constructive suggestion. We really appreciate this valid observation. Based on your insightful suggestion, we took a long time and critical revised the paper to enrich it after previous initial submission to the journal. Dear, reviewer, in the introduction part, we clearly stated that no studies have conducted in the eastern Ethiopia, and even there were very few reports from other part of the country. Further, this study was conducted in eastern part of Ethiopia, particularly, Harari Regional State, where majority of the people are not accessed to health care services. This region has boundary with Somali Regional State, where marginalized and pastoralist community are inhibited, and many of laboring mothers referred to Hiwot Fana Specialized University Hospital for better care because it is the only Referral Hospital in eastern Ethiopia. 

2. Based on the current analysis shows no significant improvement in outcomes while utilizing IoL, further analysis is merited for the use case.

Authors' Response: Great! Thanks a lot. Dear, reviewer, we really appreciate your valid concern. Initially, we followed standard methods of data collections and analysis. We also used strong statistical software (Epi-Data version-3.1) for data entry. Further, data were exported to SPSS version 24 (IBM SPSS Statistics, 2016) for further analysis. We used all statistical analysis steps required for comparison cross-sectional study (Descriptive statistics, Bi-variable analysis, Multivariable analysis). We performed regression analysis after critically considered assumption of logistic regression. Our outcome variable was “Binary outcome” and we dichotomized it accordingly. It is obvious that there would be limitation for “cross-sectional study” because no causal-association can be estimated. In this study, we used “Facility-based Cross-sectional study with internal comparison”, and we tried to increase our sample size (n=717) to have representative sample. Moreover, because of limited resource, we could not conduct, prospective longitudinal study to see the true causal-effect relationship (to estimate the true effect of predictor variables on the outcome of IOL).

2a. The paper seems to describe the effect of the parameters in isolation. The data representation, and therefore results, does not describe the effect of the combination of factors. Authors must conduct this analysis and clearly represent the significance of individual parameters and the various permutations and combinations 

Authors' Response: Thank you so much. Authors critically considered this input. Authors acknowledged your efforts. We critically considered and incorporated these raised issues. Dear reviewer, from the beginning, we followed all steps of statistical analysis as follows:

1.Questionnaire template was prepared using Epi-Dat version 3.1 for data entry

2.Double data entry was conducted by independent data clerk

3.After verification, entered data were exported to SPSS version 24 (IBM SPSS Statistics, 2016) for further analysis.

4. In SPSS analysis, imported data were cleaned and checked for outliers using (ascending and descending, simple frequency, Plot chat…etc).

5.Descript statistics were done using frequency tables, proportions and summary measures

6.Bivariable Analysis was conducted using Binary logistic regression analysis after dichotomizing of “outcome of induction as ‘Failed IOL’ and ‘Successful IOL’ ”. Thus, Failed IOL= 0 and Successful IOL= 1. Accordingly, around 12 predictors were run independently in respect with outcome of IOL.(See Table 3) 

7. Multivariable analysis was performed after all assumptions of Logistic regression were fulfilled. Here, around 8 variables with p-value < 0.25 in Bivariable analysis were considered for multivariable analysis based on selection criteria(after model fitness and multicollineaity were checked).(See Table 4)

2b. A detailed description of the control sample (patients not receiving IOL) is required. The current manuscript does not provide details on this. 

Authors' Response: Thanks a lot. Authors appreciate your insightful concern. Dear, reviewer, in this study, we used the method of “Cross-sectional study design with internal comparison” which has been conducted in point in time or “Snapshot”. We have conducted retrospective chart-review of 2 years (January 2017 to December 2018). Moreover, in its nature, cross-section study does not have comparative group like case-control (Diseased Vs Non-diseased), Cohort study (Exposed Vs Unexposed group), RCT (Placebo Vs Interventional group). Therefore, In our case, we didn’t have control group, and our study subjects were “Laboring mother who underwent IOL”. Accordingly, the charts of mothers admitted to selected Public Hospitals in Eastern Ethiopia, who underwent labor induction(January 2017 to December 2018) were retrieved. 

2c.Based on the results provided, the bishops score has a significantly higher effect on the outcome. While this may well turn out to be true, the analysis needs to be provide an analysis while holding the bishops score as a constant. This would allow a better study of the effect of each variable. Without this analysis the effect of the IOL on parameters beyond the bishops score are nebulous at best. A similar analysis should be conducted while holding each parameter constant. This would provide an exhaustive analysis that helps differentiate the output from prior work. 

Authors' Response: Thank you a lot. We appreciated your observation. We considered this deep and insightful concern. Dear, reviewer, initially, we followed all steps of statistical analysis. In the final model, multivariable analysis was performed after all assumptions of Logistic regression analysis were fulfilled. Accordingly, around 8 variables with p-value<0.25 in Bi-variable analysis were considered for multivariable analysis based on selection criteria (after model fitness and multicollineaity were checked). These variables include (maternal age, residence, parity, Bishop score, PROM, NRFHBP, Methods of induction, Birth wt). In this final model analysis, all reference categories were held constant to estimate the effect of each predictor on outcome of IOL. In the previous submission, the Bishop score was also incorporated in both Bi-variable and multivariable analysis. 

(See Table 4)

3. Authors should maintain unformity while describing results. Eg: Line 259 describes the paramenter as "more likely" based on the cut off; but the subsequent parameters are described as "less likely" at the cut off. The authors should construct the sentences to describe in one of two ways (either less or more). Switching between positive and negative effect of the cut off can create confusion for the readers. 

Authors' Response: Thank you so much. You are perfect. In fact, this is very important comment. This valid comment was addressed based on your insightful suggestion. Newly changed and corrected were highlighted with red font color in the clean revised manuscript (page , lines ).

4. The discussion section focusses more on prior work than an analysis of the results. Perhaps correlation while holding parameters constant would provide more content for discussion.

Authors' Response: Great, Thanks a lot for such implicit and critical review for our premature paper. You are perfect. We really appreciate this valid observation. Now, the authors critically reviewed this valid comment and corrected the necessary modification. The newly modified changes were highlighted with red font color in the clean revised manuscript

5. Significant sections of the paper need to be re-written to correct for sentence construction.

Authors' Response: Thank you so much. Authors critically considered this input. Authors acknowledged your efforts. We critically considered and incorporated all raised issues, comments, suggestions and concerns in this manuscript. Moreover, we thoroughly revised and edited the whole parts of our manuscript and extensively corrected all copy-editing errors in the clean revised manuscript. Authors also sent the manuscript to language expert/editor who critically reviewed, edited and corrected all language related errors made in a submitted manuscript.

6. Suggestion: Some of the tables could be converted to graphical representation for easier interpretation.

Authors' Response: Ok, Great! Thanks a lot. In fact, this is valid concern. It is obvious that graphs/bars are easier to capture the information about presenting data. In this study, we used 3 figures and 4 tables, and most Biomedical researchers recommend minimum number figures than tables. However, if is mandatory to put the data in the figure, we will modify based on your specific recommendation. Thanks!

Overall the work conducted has potential for significant impact, but a more detailed analysis is needed. In the current format the manuscript does not give any indication of differentiation from prior art. However, given the nature of the problem being addressed, I recommend the authors be give a chance to conduct a more details analysis and submit a renewed manuscript.

Authors' Response: Thank you very much. Authors thank for your countless effort to enrich our paper. We also appreciate this valuable suggestion. Dear, reviewer, we tried to address your deep concern about the analysis of the data in the previous questions. Based on your insightful comments, the authors critically reviewed, and checked the SPSS archived data to correct any bolded errors or misleading results. However, the authors didn’t found any systematic errors that may introduced unintentionally. Moreover, we took a long time, critically revised the whole manuscript, and modified previously bulky paragraphs, poorly worded, and other copy-editing errors. All newly modified/changes were highlighted with red font color with in the revised main manuscript. For further details, we submitted SPSS data (used for analysis) to the Journal as supplementary file 2(S2) with the revised submission. 

End of authors responses for Reviewer 1

B. Authors’ Response to Reviewer 2’s Comments and Suggestions

Title: Outcome of Induction and Associated Factors among Induced Labors in Public Hospitals of Harari Regional State, Eastern Ethiopia: A two Years’ Retrospective Analysis

Authors: Yimer Mohammed Beshir, Gudina Egata, Mohammed Abdurke Kure, Kedir Teji Roba

To: Reviewer 2

From: Mohammed Abdurke Kure (Corresponding Author)

Subject: Submission of Incorporated Comments and Suggestions

First and foremost, we would like to acknowledge you for your constructive and valuable comments and helpful suggestions that helped us to improve and enrich our manuscript. Here under in the table we have pointed out how authors incorporated your valuable comments, suggestions and concerns one by one. 

Reviewer’s Comments to the Authors 

Reviewer’s General Comments 

Authors' Response: Overall, thank you very much for your positive and constructive suggestions

1.Is the manuscript technically sound, and do the data support the conclusions?

Reviewer #1: Yes 

Authors' Response: Thank you very much! We appreciate your valuable response.

2. Has the statistical analysis been performed appropriately and rigorously?

Reviewer #1: Yes 

Authors' Response: Thank you very much. Authors acknowledged your countless efforts, positive and constructive suggestion! Further, we frequently revised the statistical analysis, and results to enrich the manuscript than previous its status.

3. Have the authors made all data underlying the findings in their manuscript fully available?

Reviewer #1: Yes 

Authors' Response: Thank you reviewer! We are very grateful for your positive response.

4. Is the manuscript presented in an intelligible fashion and written in standard English?

Reviewer #1: Yes 

Authors' Response: Thanks a lot! Authors also sent the manuscript to language expert/editor who critically reviewed, edited and corrected all language related errors made in a submitted manuscript.

5. Specific Review’s Comments to the Authors 

1 Abstract is written well and comprehensive

Authors' Response: Thank you very much. We would like to thank you for your appreciation and constructive suggestion. Further, we revised and enriched the paper after previous initial submission to the journal.

2. In the Introduction section, Authors may give more information about advantages and disadvantages of induction of labor (IOL) citing more literature 

Authors' Response: Thank you a lot. We accepted all your valid suggestion and concern in this paper, and correct accordingly. The newly modified part were highlighted with red-font color in the main manuscript.

3. Study design, Study population, Eligibility criteria, and Sample size calculations are well planned and presented; sampling procedure is done well with good selection of variables.

Authors' Response: Thank you very much. We would like to thank you for your appreciation and constructive suggestion. Further, we revised and enriched the paper after previous initial submission to the journal.

4 The presentation of results obtained is very clear with facts of many pregnant women having indications of pre eclampsia/eclampsia and pre-labor rupture of membrane. The state of conditions are worrying in this part Ethiopia reported by the Authors. Parity of the mothers, nullipara and multipara mothers is also taken care in this study 

Authors' Response: Thank you very much. We thank you for your appreciation and constructive suggestion. Further, we revised and enriched the paper after previous initial submission to the journal.

5. The results obtained are discussed well citing references. 

Authors' Response: Ok! Thank you so much for your countless effort to review our paper. 

6. As another parameter, in the discussion, the Authors would have touched upon socio-economic and nutritional status of pregnant women compared to those in other countries/Ethiopia. 

Authors' Response: Thank you a lot. We appreciated your observation. Now authors considered your valid concern and explicitly incorporated your input in the clean revised manuscript and highlighted with red font color.

7. In this study Authors established that success of induction of labor was relatively low in study area

Authors' Response: Thank you very much. We thank you for your appreciation and constructive suggestion. Further, we revised and enriched the paper after previous initial submission to the journal.

End of authors responses to Reviewer 2

---

## [Decision Letter · Decision Letter 1]

20 Sep 2021

PONE-D-21-06515R1Outcome of Induction and Associated Factors among Induced Labours in Public Hospitals of Harari Regional State, Eastern Ethiopia: A two Years’ Retrospective AnalysisPLOS ONE

Dear Dr. Kure,

Thank you for submitting your manuscript to PLOS ONE. After careful consideration, we feel that it has merit but does not fully meet PLOS ONE’s publication criteria as it currently stands. Therefore, we invite you to submit a revised version of the manuscript that addresses the points raised during the review process.

Please revise the manuscript considering the comments from reviewer.

We look forward to receiving your revised manuscript.

Kind regards,

Subash Chandra Gupta, Ph.D.

Academic Editor

PLOS ONE

Journal Requirements:

Additional Editor Comments (if provided):

Please revise the manuscript considering the comments from reviewer.

Reviewers' comments:

Reviewer's Responses to Questions

**Comments to the Author**

1. If the authors have adequately addressed your comments raised in a previous round of review and you feel that this manuscript is now acceptable for publication, you may indicate that here to bypass the “Comments to the Author” section, enter your conflict of interest statement in the “Confidential to Editor” section, and submit your "Accept" recommendation.

Reviewer #1: (No Response)

Reviewer #2: All comments have been addressed

2. Is the manuscript technically sound, and do the data support the conclusions?

Reviewer #1: Yes

Reviewer #2: Yes

3. Has the statistical analysis been performed appropriately and rigorously? 

Reviewer #1: Yes

Reviewer #2: Yes

4. Have the authors made all data underlying the findings in their manuscript fully available?

Reviewer #1: Yes

Reviewer #2: Yes

5. Is the manuscript presented in an intelligible fashion and written in standard English?

Reviewer #1: Yes

Reviewer #2: Yes

6. Review Comments to the Author

Reviewer #1: Most comments have been addressed. However, the merit of conducting a study using identical parameters is debatable. I understand that the socio-economic and geographic settings of this study were different, but all previous studies on the same topic with varying socio-economic studies indicated the same output as the authors. This begs the question, is the study truly unique enough to add to a body of scientific knowledge. What was the scientific hypothesis to indicate that the outcome may be different in Ethiopia. What was the rationale for expecting a different outcome? This needs to be highlighted somehwere, else it is merely a repetition of the previous studies.

Reviewer #2: Authors have corrected and modified the manuscript as per suggestions of reviewers, no more corrections

7. PLOS authors have the option to publish the peer review history of their article (what does this mean?). If published, this will include your full peer review and any attached files.

Reviewer #1: No

Reviewer #2: No

---

## [Author Response · Author response to Decision Letter 1]

3 Oct 2021

Authors’ Response to the editor’s and Reviewers’ comments and Suggestions

Manuscript ID: PONE-D-21-06515

Journal: PLOS ONE

Dear Editors and Reviewers,

Thank you so much for giving us an opportunity to submit a revised draft of our manuscript entitled “Outcome of Induction and Associated Factors among Induced Labors in Public Hospitals of Harar Town, Eastern Ethiopia: A two Years’ Retrospective Analysis” to this high visibility impact factor and peer reviewed Journal. We appreciate the time and effort that you and the reviewers dedicated to providing feedback on our manuscript. We are very grateful for the insightful comments and valuable improvements to our premature paper. We have incorporated most of the suggestions and comments made by handling editor, reviewers. All comments and suggestions are clearly stated and well addressed (a point-by-point to the reviewer’s comments and concerns). These changes are highlighted in Red font color within the clean revised manuscript. 

Authors’ Response to Editor’s Comments and Suggestions

Title: Outcome of Induction and Associated Factors among Induced Labors in Public Hospitals of Harari Regional State, Eastern Ethiopia: A two Years’ Retrospective Analysis

Authors: Yimer Mohammed Beshir, Gudina Egata, Mohammed Abdurke Kure, Kedir Teji Roba, 

To: Handling Editor(s)

From: Mohammed Abdurke Kure (Corresponding Author)

Subject: Submission of Incorporated Comments and Suggestions

First, we thank you for your constructive comments and helpful suggestions that helped us to improve and enrich our manuscript. Here under in the table below, we have pointed out how authors incorporated your valuable comments, suggestions and concerns one by one. 

Editor’s Comments to the Authors 

Editor’s General Comments and Suggestions 

Authors' response: Overall, thank you so much for cooperation to handle our manuscript. Handling paper is really needs dedication and strong commitment. Thanks a lot!

1.Thank you for submitting your manuscript to PLOS ONE. After careful consideration, we feel that it has merit but does not fully meet PLOS ONE’s publication criteria as it currently stands. Therefore, we invite you to submit a revised version of the manuscript that addresses the points raised during the review process. 

Authors' response: Thank you so much. We are very grateful for giving us an opportunity to submit our revised manuscript to such legitimate and high visibility impact factor Journal (PLOS ONE). Thank you editor.

Academic Editor’s Specific Comments (Journal Requirements) 

Authors' response: Thank you so much. You are perfect. These are very important suggestions. Even it is the authors mandatory to critically review references issue (Both in text-citation and Bibliography) to avoid any inconsistence and wrong citations across the document. Now, the authors addressed this critical issues based on your valuable suggestions. Newly changed and corrected References were highlighted with red font color in the clean revised manuscript.

End of authors’ responses for Handling Editor(s)

Authors’ Response to Reviewer 1’s Comments and Suggestions

Title: Outcome of Induction and Associated Factors among Induced Labors in Public Hospitals of Harari Regional State, Eastern Ethiopia: A two Years’ Retrospective Analysis

Authors: Yimer Mohammed Beshir, Gudina Egata, Mohammed Abdurke Kure, Kedir Teji Roba

To: Reviewer 1

From: Mohammed Abdurke Kure (Corresponding Author)

Subject: Submission of 3rd Round Incorporated Comments and Suggestions

Above all, we would like to thank you for your constructive and valuable comments and helpful suggestions that helped us to improve and enrich our premature manuscript. Here under in the table below, we have pointed out how authors incorporated your valuable comments, suggestions and concerns one-by-one. 

Reviewer’s Comments to the Authors Authors’ Responses to Reviewer’s comments

Reviewer’s General Comments Overall, thank you very much for your positive and constructive suggestions

1. If the authors have adequately addressed your comments raised in a previous round of review and you feel that this manuscript is now acceptable for publication, you may indicate that here to bypass the “Comments to the Author” section, enter your conflict of interest statement in the “Confidential to Editor” section, and submit your "Accept" recommendation.

Reviewer #1: (No Response) 

Authors' response: Ok, great! Thank you so much. We revised and enriched the paper after previous 2nd round submission to the journal. Authors critically discussed on this reviewers suggestion (not endorsed for previous addressed comments) to the Journal’s query to enrich our paper to be scientifically sound, and we took a long time, implicitly revised for the reviewers concerns. Thus, now we clearly stated and addressed the this reviewer’s deep concern(particularly, the issue of novelty and duplication), in the question number 5 below(next page). Thank you a lot, reviewer.

2. Is the manuscript technically sound, and do the data support the conclusions?

Reviewer #1: Yes 

Authors' response: Thank you. Authors appreciate your positive response. 

3. Has the statistical analysis been performed appropriately and rigorously?

Reviewer #1: Yes 

Authors' response: Thank you so much. Authors acknowledge you for your countless effort. We also appreciate this valuable response. 

4. Have the authors made all data underlying the findings in their manuscript fully available?

Reviewer #1: Yes 

Authors' response: Great! Dear reviewer, we thank you again for constructive suggestion.

5. Is the manuscript presented in an intelligible fashion and written in standard English?

Reviewer #1: Yes 

Authors' response: Thank you very much. Even after revised 2nd submission, we corrected all copy-editing errors in the clean revised manuscript after previous submission. 

5. Specific Review’s Comments to the Authors 

1. Most comments have been addressed. However, the merit of conducting a study using identical parameters is debatable. I understand that the socio-economic and geographic settings of this study were different, but all previous studies on the same topic with varying socio-economic studies indicated the same output as the authors. This begs the question, is the study truly unique enough to add to a body of scientific knowledge. What was the scientific hypothesis to indicate that the outcome may be different in Ethiopia? What was the rationale for expecting a different outcome? This needs to be highlighted somewhere, else it is merely a repetition of the previous studies. 

Authors' response: Thank you very much. We would like to thank you for your appreciation and constructive suggestion. 

Coming to the specific concern: Really, this is an important intellectual suggestion. Dear, reviewer, authors really appreciate your implicit and deep concern about the novelty and repetition of this research paper. In this regard, we tried to indicate in the 2nd round submission of the revised manuscript and review letter response. 

Dear, reviewer, in the introduction part of this paper, we clearly stated that no studies have been conducted in the eastern part of Ethiopia, and even there were very few reports from other part of the country. Further, this study was conducted in eastern part of Ethiopia, particularly, Harari Regional State, where majority of the people are less accessed to health care services. Moreover, this Harari Reginal State has boundary with Somali Regional State, where marginalized and pastoralist community are inhibited, and many of laboring mothers referred to Hiwot Fana Specialized University Hospital for better care because it is the only Referral Hospital in eastern Ethiopia. Furthermore, in Ethiopia, eastern part of Ethiopia is one of the most neglected area of the country, particularly, Harari regional state which is located at periphery( at 526Km distance from capital Addis Aba) of the country, and has a boundary with Somali regional, where highly marginalized community(Pastoralists and semi pastoralist) communities are largely inhibited and seeking healthcare services from Harari regional state, Eastern Ethiopia. 

- In Ethiopia, although few studies have been conducted in the last five years, almost all previous researchers were selective to central and Northern parts(Addis Ababa, Amhara, and Tigray regions), neglecting other regions of the country, particularly Afar, Harari, and Somali regions. 

Further, the authors added researcher gaps that indicate the reasons why this study has been conducted in Eastern part of Ethiopia. The newly modified changes were highlighted within the clean revised manuscript on page (page 5, lines 87-94). Thank you once again. 

Dear, reviewer, really publication is learning forum because we learned a lot from your insightful comments and suggestions throughout the review process our paper. 

End of authors responses for Reviewer 1

Authors’ Response to Reviewer 2’s Comments and Suggestions

Title: Outcome of Induction and Associated Factors among Induced Labors in Public Hospitals of Harari Regional State, Eastern Ethiopia: A two Years’ Retrospective Analysis

Authors: Yimer Mohammed Beshir, Gudina Egata, Mohammed Abdurke Kure, Kedir Teji Roba

To: Reviewer 2

From: Mohammed Abdurke Kure (Corresponding Author)

Subject: Submission of Incorporated Comments and Suggestions

First and foremost, we would like to acknowledge you for your constructive and valuable comments and helpful suggestions that helped us to improve and enrich our manuscript. Here under in the table we have pointed out how authors incorporated your valuable comments, suggestions and concerns one by one. 

Reviewer’s Comments to the Authors

Reviewer’s General Comments 

Authors' response: Overall, thank you very much for your positive and constructive suggestions

1.If the authors have adequately addressed your comments raised in a previous round of review and you feel that this manuscript is now acceptable for publication, you may indicate that here to bypass the “Comments to the Author” section, enter your conflict of interest statement in the “Confidential to Editor” section, and submit your "Accept" recommendation.

Reviewer #2: All comments have been addressed Great! 

Authors' response: We are very grateful for your unquantified effort throughout the review process of our manuscript. Thank you so much. 

2.Is the manuscript technically sound, and do the data support the conclusions?

Reviewer #2: Yes 

Authors' response: Thank you very much! We appreciate your valuable response and constructive suggestion. 

3. Has the statistical analysis been performed appropriately and rigorously?

Reviewer #2: Yes 

Authors' response: Thanks a lot. Authors acknowledged your countless efforts, positive and constructive suggestion! 

4. Have the authors made all data underlying the findings in their manuscript fully available?

Reviewer #2: Yes 

Authors' response: Thank you reviewer! We are very grateful for your appreciation and constructive intellects. 

5. Is the manuscript presented in an intelligible fashion and written in standard English?

Reviewer #2: Yes 

Authors' response: Thank you so much! Authors also sent the manuscript to language expert/editor who critically reviewed, edited and corrected all language related errors made in a submitted manuscript.

5. Specific Review’s Comments to the Authors

Reviewer #2: Authors have corrected and modified the manuscript as per suggestions of reviewers, no more corrections

 Thank you very much. We would like to thank you for your appreciation and constructive suggestion. Further, we revised and enriched the paper after previous 2nd round submission to the PLOS ONE Journal. 

In conclusion:

Really, the publication is leaning forum. Overall, we learned a lot from your comments and suggestion throughout the review process of this manuscript. Thank you so much once again.

End of authors responses to Reviewer 2

---

## [Decision Letter · Decision Letter 2]

26 Oct 2021

Outcome of Induction and Associated Factors among Induced Labours in Public Hospitals of Harari Regional State, Eastern Ethiopia: A two Years’ Retrospective Analysis

PONE-D-21-06515R2

Dear Dr. Kure,

We’re pleased to inform you that your manuscript has been judged scientifically suitable for publication and will be formally accepted for publication once it meets all outstanding technical requirements.

Kind regards,

Subash Chandra Gupta, Ph.D.

Academic Editor

PLOS ONE

Additional Editor Comments (optional):

Reviewers' comments:

Reviewer's Responses to Questions

**Comments to the Author**

1. If the authors have adequately addressed your comments raised in a previous round of review and you feel that this manuscript is now acceptable for publication, you may indicate that here to bypass the “Comments to the Author” section, enter your conflict of interest statement in the “Confidential to Editor” section, and submit your "Accept" recommendation.

Reviewer #1: All comments have been addressed

Reviewer #2: All comments have been addressed

2. Is the manuscript technically sound, and do the data support the conclusions?

Reviewer #1: Yes

Reviewer #2: Yes

3. Has the statistical analysis been performed appropriately and rigorously? 

Reviewer #1: Yes

Reviewer #2: Yes

4. Have the authors made all data underlying the findings in their manuscript fully available?

Reviewer #1: Yes

Reviewer #2: Yes

5. Is the manuscript presented in an intelligible fashion and written in standard English?

Reviewer #1: Yes

Reviewer #2: Yes

6. Review Comments to the Author

Reviewer #1: The study is unique for the particular for the body of knowledge being built in Ethiopoa. It is my sincere hope that the work presented here will enable improved health outcomes in Ethiopia

Reviewer #2: Authors have made corrections and added relevant text as suggested by reviewers, no more corrections

7. PLOS authors have the option to publish the peer review history of their article (what does this mean?). If published, this will include your full peer review and any attached files.

Reviewer #1: No

Reviewer #2: No

---

## [Editor Report · Acceptance letter]

29 Oct 2021

PONE-D-21-06515R2 

Outcome of Induction and Associated Factors among Induced Labours in Public Hospitals of Harari Regional State, Eastern Ethiopia: A two Years’ Retrospective Analysis 

Dear Dr. Kure:

I'm pleased to inform you that your manuscript has been deemed suitable for publication in PLOS ONE. Congratulations! Your manuscript is now with our production department. 

Kind regards, 

on behalf of

Dr. Subash Chandra Gupta 

Academic Editor

PLOS ONE